# Targeting Protein Kinase C for Cancer Therapy

**DOI:** 10.3390/cancers14051104

**Published:** 2022-02-22

**Authors:** Sijia He, Qi Li, Qian Huang, Jin Cheng

**Affiliations:** Cancer Center, Shanghai General Hospital, Shanghai Jiao Tong University School of Medicine, Shanghai 201620, China; he-si-jia@163.com (S.H.); leeqi2001@hotmail.com (Q.L.); huangqian_sjtu@163.com (Q.H.)

**Keywords:** protein kinase C, cancer, cell death, targeted therapy

## Abstract

**Simple Summary:**

The protein kinase C (PKC) family belongs to serine-threonine kinases and consists of several subtypes. Increasing evidence suggests that PKCs are critical players in carcinogenesis. Interestingly, PKCs exert both promotive and suppressive effects on tumor cell growth and metastasis, which have attracted immense attention. Herein, we systematically review the current advances in the structure, regulation and biological functions of PKCs, especially the relationship of PKCs with anti-cancer therapy-induced cell death, including the current knowledge of PKCs function in tumor metabolism and microenvironment. Moreover, we discuss the potential role of PKCs as a target for therapeutic intervention in cancer from basic research and clinical trials.

**Abstract:**

Protein kinase C (PKC) isoforms, a group of serine-threonine kinases, are important regulators in carcinogenesis. Numerous studies have demonstrated that PKC isoforms exert both positive and negative effects on cancer cell demise. In this review, we systematically summarize the current findings on the architecture, activity regulation and biological functions of PKCs, especially their relationship with anti-cancer therapy-induced cell death. Additionally, we elaborate on current knowledge of the effects of PKCs on tumor metabolism and microenvironment, which have gained increasing attention in oncology-related areas. Furthermore, we underscore the basic experimental and clinical implications of PKCs as a target for cancer therapy to evaluate their therapeutic benefits and potential applications.

## 1. Introduction

Protein kinase C (PKC) isoenzymes are a prototypical class of serine/threonine kinases that are conserved throughout evolution and widely expressed in almost all cell types. They consist of domain permutations in their N-terminal regulatory regions linked by variable sequences to a highly conserved C-terminal kinase domain. The biological role of PKC isoenzymes has been intensely studied in both various physiological and pathological conditions. In addition, PKC isoenzymes are implicated in multiple signal transduction pathways that turn environment cues into cellular actions. The PKC signaling pathway has critical roles in regulating gene expression, as well as cell proliferation, differentiation, migration, survival and death. Inactivation or dysregulation of PKC isoenzymes may cause different pathologies, such as allergy [1], inflammatory diseases [2], diabetes [3] and autoimmune diseases [4,5]. Recent scientific breakthroughs and technological advancements have improved our understanding of the functional roles of PKCs in cancer development and progression. Moreover, PKC enzymes can serve as promising targets for cancer treatment [6]. In this review, we focus on the regulatory activities of PKC isoforms and discuss their involvement in cancer progression. Furthermore, we provide a summary of clinical trials with PKC inhibitors in cancers and highlight the latest development of PKCs as targeted anti-cancer therapies.

## 2. The Structural Features and Activation of PKC Isoforms

PKC isoforms were first identified and found to be activated by proteolysis in 1977 [7]. These isoforms are now known to be critical mediators in signal transduction pathways, which regulate a set of biological functions such as cell proliferation, differentiation, death and tumorigenesis.

The PKC family members are classified into three subfamilies according to their cofactor dependence: (i) conventional or classical PKC isozymes (cPKCs), including PKCα, PKCβ1, PKCβ2 and PKCγ, which are calcium-dependent and activated by both diacylglycerol (DAG) and phosphatidylserine (PS); (ii) novel PKC isozymes (nPKCs), including PKCδ, PKCε, PKCη and PKCθ, which are calcium-independent and regulated by DAG and PS; and (iii) atypical PKC isozymes (aPKCs), including PKCζ and PKCλ/ι, which are calcium-independent and do not require DAG for activation, although PS can regulate their activity.

Members of the PKC family share a common architecture of an N-terminal regulatory moiety (approximately 20–40 kDa) and a C-terminal kinase domain (approximately 45 kDa) [8], composed of conserved regions (C1–C4) interspersed with variable regions (V0–V5), which are of lower homology. The C1 domain, a cysteine-rich motif, is duplicated in most PKC isoenzymes and forms the DAG/phorbol ester binding site, which is preceded by an autoinhibitory pseudosubstrate sequence. The C2 domain contains a calcium-binding site or recognition site for acidic lipids. The C3 and C4 domains constitute the ATP- and substrate-binding lobes of the kinase core. There is a hinge region between the regulatory and catalytic domains that becomes proteolytically labile when the enzyme is membrane-bound [8,9] (Figure 1). 

PKC is initially considered an unphosphorylated form (immature) associated with the membrane [10] and under acute structural and spatial regulation including its phosphorylation state, conformation and subcellular location [11,12]. Activation of PKC isoforms strictly relies on a posttranslational maturation process (Figure 2). Thus, when processed by three ordered phosphorylation (activation loop phosphorylation, turn motif phosphorylation, and hydrophobic motif phosphorylation), PKCs will expose their pseudosubstrate and position at the specific intracellular location, transducing signaling [11,12,13]. The first step of this maturation process is regulated by 3-phosphoinositide-dependent protein kinase-1 (PDK1), which phosphorylates threonine residues at the activation loop in the C4 region of PKC isoforms [14]. Such phosphorylation then leads to the autophosphorylation of the “turn” and “hydrophobic” motifs within the C4 region and stabilizes these isoenzymes (maturation) [10]. At this point, the mature form of PKC is prepared for activation. Its inactive form (“folded conformation”) resides predominantly in the cytosol, with an interaction between the C-terminal domain and the N-terminal pseudosubstrate motif [10]. Under physiological conditions, PKC can be activated by interacting with extracellular agonists such as hormones, cytokines and growth factors, which sequentially translocates to different subcellular compartments, thus conferring specific substrate phosphorylation and distinct cellular functions [15]. The translocation of PKC is regulated by anchoring/scaffolding proteins to ensure proper spatio-temporal distribution of PKC isoforms. Receptors for activated C kinase (RACKs) are intracellular scaffolding proteins that bind to individual PKC isoforms following their activation and provide anchorage of each isoform next to its physiological substrates [16]. Subsequently, the translocated PKC phosphorylates specific substrates such as membrane-associated phospholipase C (PLC), which hydrolyzes the membrane phospholipid phosphatidylinositol 4,5-bisphosphate (PIP2) to generate DAG and inositol 1,4,5-trisphosphate (IP3). IP3 then interacts with the inositol trisphosphate receptor (InsP3R) and triggers the rapid release of Ca^2+^ from the intracellular store to the endoplasmic reticulum (ER). DAG is phosphorylated by diacylglycerol kinases (DGK) and converts to phosphatidic acid (PA) in a reaction that terminates the PKC-regulated signals [17]. The reversal of PKC activity is regulated by dephosphorylation, ubiquitination, and caveolin-dependent endocytosis [18,19,20]. In summary, PKC signaling is initiated by phosphorylation, pseudosubstrate exposure and subcellular localization, which ultimately leads to a specific cellular response.

## 3. PKCs in Cancer

### 3.1. Differential Expression and Mutation of PKCs in Cancers

The PKC isoforms show variable expression profiles during cancer progression depending on cell types. For example, in the transitional cell carcinoma of the bladder, PKCα and β2 immunostains were intense in both normal urothelium and all grade tumor tissues. At the same time, PKCβ1 and δ immunostains were intense in normal urothelium and low-grade tumors, but weak in high-grade tumors [21]. In human liver cancer, the level of membrane-bound PKCα was significantly lower in cancer tissues than in adjacent normal tissues. The levels of PKCδ and λ/ι in both cytosolic and membrane fractions were significantly higher in cancer tissues than in adjacent normal tissues [22]. Moreover, Western blot results showed that the expression levels of PKC isoforms (α and λ/ι) were remarkably higher in cancerous tissues than in normal breast tissues [23]. The expression of PKCε was correlated with high histologic grade and poor disease-free and overall survival in breast cancer patients [24]. PKC isoforms (α, β, ε and η) were found to be highly expressed in prostate cancer compared with benign prostatic hyperplasia [25]. 

The PKC family contains nine genes and is frequently mutated in human cancer [26]. Notably, there are 20–25% PKC mutations verified in melanoma, colorectal cancer and lung squamous cell carcinoma, while less than 5% in ovarian cancer, glioblastoma and breast cancer [27]. PKCα mutation (D294G in the hinge region) was the first reported cancer-associated mutation in PKC, which separated the regulatory and catalytic moieties and abolished the F-actin dependent PKCα accumulation at the cell-cell contacts [28,29]. It has been demonstrated that nPKC isoform mutations are most common in gastrointestinal cancer, with a lower mutation burden than melanoma and lung cancer [26]. Although there are about 8% of PKC mutations confirmed in human cancers, which are heterozygous with an allele frequency varying from 0.05 to 0.67 for the mutations characterized, most of them are considered to have a loss of function rather than activation [26]. For instance, the PKCε R162H pseudosubstrate mutation displayed agonist-stimulated and basal activity limitations [26]. The PKCβ P619Q C-terminal tail mutation prevented the phosphorylation of PKC, which was also a type of loss-of-function mutation [30]. Generally, PKC loss-of-function mutations impeded second-messenger binding through mutations in the C1 or C2 domain, and impaired phosphorylation or catalysis via mutations in the kinase domain. However, gain-of-function mutations are rare in PKC isoforms and have been most commonly reported in Alzheimer’s disease. For example, PKCα M489V mutation permitted synaptic defects in patients with Alzheimer’s disease via increasing signaling output and evasion of kinase degradation [31]. Additionally, PKC fusions have been identified in some rare cancers. *SLC44A1* (solute carrier family 44)-*PRKCA* (PKCα) fusion oncogene was detected in papillary glioneuronal tumor with the rearrangement of chromosomes 9 and 17, thus leading to constitutive PKCα activation [32]. The benign fibrous histiocytomas (BFHs) have been found to possess gene fusions encoding membrane-associated proteins (podoplanin, CD63 and LAMTOR1) and PKC isoforms (PKCα, PKCβ and PKCδ), indicating that BFHs support the pathogenesis of neoplasia [33]. Despite these rare cases, deregulation of PKC activity is generally not involved in genomic alterations.

### 3.2. The Role of PKC in Anti-cancer Treatment-Induced Cell Death

Over the last decade, it has become apparent that anti-cancer therapy causes tumor cells to die indirectly or directly. Cell death manifests with macroscopic morphological alterations and has been historically classified into three different forms: apoptosis, autophagy, and necrosis. Since 2005, the Nomenclature Committee on Cell Death (NCDD) has focused on the precise definition and consensus criteria of cell death modalities, which has recently subdivided cell death into over ten interconnected subclasses, such as extrinsic apoptosis, intrinsic apoptosis, autophagy-dependent cell death (ADCD), immunogenic cell death (ICD) and necroptosis [34]. Balancing cell death and survival are crucial for preventing cancer and improving therapy efficiency.

#### 3.2.1. PKCs and Apoptosis

Apoptosis is a programmed cell death process mediated by intrinsic and extrinsic signaling pathways. Promoting the effective elimination of cancer cells by apoptosis has been a mainstay and goal of clinical oncology during the last decade. PKC isoforms appear to positively and negatively regulate cancer cell apoptosis. In general, PKCα protects cancer cells against apoptosis. PKCα has been shown to phosphorylate B-cell lymphoma 2 (Bcl-2) and dephosphorylate Akt, which triggers a survival response [35]. The oxygen radicals induced by irradiation have been substantiated to activate PKC in a direct way [36]. Following irradiation, PKCα translocates to biomembranes through lipid peroxidation and is activated, thus negatively regulating radiation-induced apoptosis, whereby Bcl-2, Raf-1 and HSP70 may be key molecules [36]. Inhibition of PKC decreases the initiation and elongation activity of the HSP70 gene, which protects cells from undergoing apoptosis after heat shock [37]. Tumor necrosis factor receptor-associated factor 1 (TRAF1), an anti-apoptotic component of the intracellular signaling pathway of the tumor necrosis factor receptor (TNFR) family, can be stimulated by PKC activation via Raf-1/extracellular signal-regulated kinase (ERK)/nuclear factor-κB (NF-κB)-dependent pathway [38]. Overexpression of PKCα rendered cancer cells resistant to irradiation. Downregulation of PKCα-induced p53-dependent apoptosis via activation of insulin-like growth factor-binding protein-3 (IGFBP3) [36,39]. Enhancement of PKCε expression confers lung cancer cells a significant resistance to chemotherapeutic drugs induced apoptosis via suppressing cytochrome c release from the mitochondria, activation of caspase-3, and cleavage of poly (ADP-ribose) polymerase (PARP) [40]. PKCη also appears to play an important role in apoptosis regulation, and its downregulation triggers tumor necrosis factor-related apoptosis-inducing ligand (TRAIL)-induced apoptosis [41]. However, PKCβ [42], δ [43,44], ζ [45] and θ [46,47] are supposed to positively regulate apoptosis. Cisplatin promotes the expression of PKCβ in cervical cancer cells, and inhibition of PKCβ reduces cisplatin-induced cancer cell apoptosis [42]. PKCδ functions downstream of the p53 response and upstream of c-Jun N-terminal kinase (JNK) activation, leading to cytochrome C release from mitochondria and caspase-3 cleavage [43]. Rad9, DNA-dependent protein kinase (DNA-PK), Mcl-1 and p73 are also targets of PKCδ to regulate apoptosis. PKCδ is required for the formation of Rad9-Hus1-Rad1 complex and binding to Bcl-2, and inhibition of PKCδ attenuates Rad9-induced apoptosis [44]. Irradiation initiates c-Abl/PKCδ/Rac1/p38 cascades and leads to the oligomerization of Bax and Bak, which results in the release of cytochrome C [48]. It has been demonstrated that PKCα might not contribute to the progression of kidney carcinoma since its inhibition does not affect kidney cancer cell apoptosis [49]. Additionally, in some reports, PKC activation by oxidative stress and irradiation is not always accompanied by translocation [50]. The activation of PKC at an early stage after irradiation promoted anti-apoptosis signaling, but not through translocation [50]. PKCζ is involved in indomethacin-induced apoptosis via activating the p38-DRP1 (dynamin-related protein 1) pathway [45]. PKCθ could be cleaved by caspase-3 and apoptotic cell lysates at the DEVD354/K site [46]. However, PKCζ is one of the molecules involved in the phosphatidylinositol 3-kinase pathway transducing cell survival signals, knockdown of PKCζ sensitizes colon cancer cells to TRAIL-induced apoptosis [51]. Moreover, a high expression of PKCθ is sufficient to inhibit paclitaxel-induced apoptosis. Knockdown of PKCθ expression also enhanced chemotherapy-induced apoptosis in triple-negative breast cancer cells via regulating Bim [52]. 

#### 3.2.2. PKCs and Autophagy

Autophagy is a dynamic process in which the cells recycle their components to survive under stressful or starvation conditions [53]. The role of autophagy in cancer is context-dependent. Autophagy prevents cell damage that can lead to cancer initiation, thus playing a cytoprotective role in cancer prevention. However, once cancer occurs, the cancer cells may undergo autophagy to survive in the hostile tumor environment. In this setting, aggressive cancers can be addicted to autophagy for survival. Multiple signaling pathways are involved in the process of autophagy, such as mitogen-activated protein kinases (MAPK)/JNK, phosphatidylinositol-3-kinase (PI3K)/Akt/mammalian target of rapamycin (mTOR), and PKC-related signaling pathway [54,55,56]. PKC enzymes contribute to cellular homeostasis maintenance by regulating the autophagy of cellular components. As autophagy regulators, PKC enzymes mediate the function of other autophagy-related proteins via phosphorylation of serine and threonine amino acid residues. Silencing of PKCε by siRNA suppressed starvation-induced autophagy in breast cancer cells. Overexpression of PKCε activated mTORC1 and increased the expression of LC3-I, LC3-II and p62 in MCF-7 cells [53]. Overexpression of PKCβ sensitized HeLa cells to chemotherapy via autophagy enhancement [42]. However, the role of PKC enzymes in cancer cell autophagy is contradictory. Knockdown of PKCα impaired LC3-II accumulation and autophagic flux in palmitic acid-treated HepG2 cells [57]. PKCα activator positively regulated tri-ortho-cresyl phosphate (TOCP)-induced autophagy by enhancing the protein expression level of LC3 and P62 in neuroblastoma SK-N-SH cells [58]. Nevertheless, one study reported that PKCα suppressed autophagy by regulating miR-129-2, which directly targeted peroxisome proliferator-activated receptor 1 alpha (PGC-1α), a positive regulator of autophagy [59]. PKCδ has been shown to promote autophagy by increasing the formation of autophagosomes and blocking autophagic flux [60]. Rottlerin, a PKCδ inhibitor, could trigger early autophagy by phosphorylating AMP-activated protein kinase (AMPK) at Thr-172 or suppressing Akt/mTOR signaling pathway in human breast cancer stem cells [61]. These paradoxical results may be explained by the fact that PKC enzymes exhibit different intracellular distribution in different cancer cells and they translocate to different organelles upon activation.

#### 3.2.3. PKCs and Necrosis

In the last decade, necrosis has been identified as a programmed cell death process with cellular swelling and cytoplasmic membrane rupture. Previous research has shown that the modulation of PKC and AMPK signaling pathways can affect H_2_O_2_-induced necrosis [62]. Under H_2_O_2_ treatment, aPKC isoforms were activated and subsequently inhibited the AMPK signaling pathway, which was crucial for necrosis promotion [62]. Necroptosis, a kind of caspase-independent programmed necrosis, has been recently investigated in cancer cells after treatment with anticancer agents. Ascorbic acid induces intracellular reactive oxygen species (ROS) generation and PKC activation, especially PKCα and PKCβ2 enzymes, which in turn affects the release of calcium into the cytosol and eventually leads to Hep2 cell necrosis [63]. The mechanisms of the cell death mode switch may point out a new way for the treatment of necrosis-mediated tumor growth. It has been suggested that the necrosis-to-apoptosis switch in solid tumors is controlled by PKC- ERK1/2 pathway, which regulates superoxide dismutase (SOD) and suppresses ROS production [64].

#### 3.2.4. PKCs and Other Cell Death Modalities

Cellular senescence is a permanent state of cell cycle arrest that prevents the cells from acquiring unnecessary damage from different types of stressors, in conjunction with the secretion of the senescence-associated secretory phenotype. In cancer, senescence is considered a potent barrier for preventing tumorigenesis [65]. Studies have shown that PKC enzymes play an important role in senescence. Inhibition of cPKC or aPKC triggered the activation of senescence markers in colon cancer cells via the Akt-FoxO3a-ROS-p53-p21 pathway [66]. Depletion of PKCλ/ι increased the number of senescent cells in PKCλ/ι-overexpressing breast cancer cells and glioblastoma cells, in which PKCλ/ι was activated by loss of *PTEN* [67]. PKCη promoted senescence induced by oxidative stress and DNA damage through upregulating the expression of p21^Cip1^ and p27^Kip1^. In addition, PKCη was demonstrated to enhance the transcription and secretion of interleukin-6 (IL-6) for a positive loop of senescence reinforcement [68]. However, PKCη depletion could also promote senescence by enhancing p27 [69]. Ferroptosis is an iron-dependent form of cell death characterized by ROS accumulation and lipid peroxidation. Erastin, a well-known ferroptosis inducer, triggered rhabdomyosarcoma cell death in a dose-dependent manner, which was reduced by PKCα and β inhibition. Genetic knockdown of PKCα significantly prevented RMS cells from erastin-induced cell death [70]. However, there is a lack of research on the relationship between PKC and other emerging cell death modalities. 

### 3.3. PKCs and Epithelial-Mesenchymal Transition (EMT)

Epithelial-mesenchymal transition (EMT) is a process in which epithelial cancer cells lose their epithelial differentiated characteristics and gain mesenchymal characteristics with spindle-shaped morphology to enhance their migratory and invasive potential. During embryonic development, EMT is essential for organ formation, whereby the cells disseminate from the primary site and migrate to a distant site [71]. However, during cancer progression, this process was hijacked by cancer cells for metastasis to different organs. Five main pathways have been demonstrated to trigger the EMT process and EMT-like phenotypes: (i) tyrosine kinase receptors including epidermal growth factor (EGF), fibroblast growth factor (FGF), insulin-like growth factor (IGF), plate-derived growth factor (PDGF) and hepatocyte growth factor (HGF); (ii) integrins; (iii) Wnt pathway; (iv) NF-κB pathway; and (v) transforming growth factor β (TGF-β) pathway [72]. EGF is known to affect adherens junctions and disrupt cell-cell adhesion in cancer. It has been reported that EGF can phosphorylate PKCδ at tyrosine 311, which is necessary for its binding to E-cadherin. Knockdown of PKCδ with siRNA or impeding PKCδ phosphorylation by overexpression of a mutant repressed EFG-induced cell scattering [73]. In recent years, integrin trafficking has shown its importance in cell migration during cancer metastasis. PKC isoforms appear to function as a vital center in regulating the vesicular pathways of integrins. The trafficking of αvβ3, α2β1 and β1 integrins all hinged on PKC activity [74]. Many studies have implicated the crosstalk between the Wnt and PKC signaling pathways. PKCθ positively mediated the canonical Wnt signaling pathway, whereas PKCη and δ played negative roles in this signaling pathway [75]. Activation of NF-κB is associated with cancer cell migration and metastasis. TPA (12-*O*-tetradecanoylphorbol-13-acetate) has been reported to activate PKCα/ERK/NF-κB-dependent matrix metalloproteinase 9 (MMP-9) expression and elicit glioblastoma cell migration, which can be prevented through downregulating PKCα expression [76]. TGF-β signaling is an important regulator of EMT. In conjunction with the canonical TGF-β/Smad signaling pathway that promotes EMT via altering transcriptional responses, another TGF-β signaling pathway was verified where Par6 (an adaptor molecule for the polarity complex) functioned as a binding partner and substrate of TGF-β receptors. aPKC was identified as a crucial component in this TGF-β-activated phosphorylation of the Par6 pathway, and its inhibition decreased TGF-β-induced EMT and migration of non-small cell lung cancer (NSCLC) cells [77]. Additionally, overexpression of PKCε induced a mesenchymal phenotype of MCF-10A cells and promoted metastasis, which was partially reversed by PKCε knockdown [78]. PKCα inhibitor blocked Wnt5a-enhanced EMT in ovarian cancer [79]. Specificity protein 1 was directly phosphorylated by PKCλ/ι on Ser59, which induced cholangiocarcinoma cells with EMT-like features via binding to Snail promoter [80]. Furthermore, knockdown of apoptosis-stimulating of p53 protein 2 (ASPP2) promoted EMT in gallbladder cancer cells through the glioma-associated oncogene homolog 1 (PKCλ/ι/GLI1) pathway [81].

### 3.4. PKCs and Tumor Microenvironment (TME)

The tumor microenvironment (TME) consists of stromal cells, fibroblasts, endothelial cells, and immune cells, which can be functionally sculpted by cancer cells to foster cancer development and progression. Many PKC enzymes are shown to regulate immune cell function, which is germane to the definition of promotion or suppression of cancer cell growth. PKCα is necessary for T cell-dependent interferon (IFN)-gamma production and B cell IgG2a/2b class-switching [82]. PKCβ is essential in B cell receptor (BCR) signaling, while dispensable for T cell receptor (TCR) signaling [83]. PKCβ has been found to promote tumorigenesis in mouse mammary tumor virus-polyoma middle T-antigen (*MMTV-PyMT*) breast cancer model via mediating TME [84]. PKCε is implicated in T cell activation and differentiation [85]. It has been well documented that PKCθ is involved in T-cell activation, differentiation and proliferation, thus negatively regulating the suppressive function of Treg cells [86]. Further analysis demonstrated that PKCλ/ι expression was significantly associated with poorly differentiated hepatocellular carcinoma and late recurrence in patients with liver cancer. The deficiency of PKCλ/ι in non-tumorous liver tissue generated a pro-tumorigenic liver microenvironment, indicating this kinase is a tumor suppressor in liver cancer [56]. Cytotoxic T lymphocyte-associated protein 4 (CTLA-4), a negative regulator of T cell-mediated immune responses, could physically interact with phosphorylated PKCη and colocalized in regulatory T cells formed immunological synapse with antigen-presenting cells. PKCη-deficient Treg cells displayed defective suppression of tumor immunity via CTLA-4-PKCη signaling axis [87]. Angiogenesis is important for tumor growth and metastasis by enhancing nutrients and oxygen supply, which is modulated by different molecules and cells in TME [88]. Vascular endothelial growth factor (VEGF) plays a vital role in angiogenesis. It has been reported that VEGF binds to VEGF receptors and subsequently phosphorylates PLC-γ, and activates PKCβ, which in turn promotes MAPK signaling pathway activation and tumor angiogenesis and growth in breast cancer [89]. It has been demonstrated that PKCβ2 is activated in prostate cancer during angiogenesis, which can be impeded by PKCβ2 inhibitor [90]. Our previous work revealed that caspase-3 cleaved PKCδ and actively secreted VEGF-A into TME and elicit potent growth-stimulating signals to promote tumor repopulation [91].

### 3.5. PKCs and Tumor Metabolism

Aberrant metabolism is a major hallmark of cancer and directly affects tumor signaling and cellular reactions [92]. In the 1920s, Otto Warburg first proposed the concept that, unlike normal cells, cancer cells obtained ATP through glycolysis, even in oxygen-abundant environments, which was commonly known as the Warburg effect [93]. The maintenance of stable glucose metabolism is important for cancer cell growth and development. To better respond to nutrient stress when glucose is limited, tumor cells can metabolize glutamine, which contributes to mainly every core metabolic task of proliferating tumor cells [94]. It has been demonstrated that bone-derived breast cancer cells show a low expression level of PKCζ compared to the parental population, with low glycolytic reserve and glucose uptake while enhancing glutamine metabolism. These findings suggest that breast cancer cells reprogram metabolic pathways to respond to altered nutrient requirements in the bone microenvironment via upregulation of serine biosynthesis pathway genes and downregulation of PKCζ [95]. Additionally, under glucose deprivation conditions, PKCζ deficiency induces the metabolism to reprogram by increasing glutamine utilization through the serine biosynthetic pathway, where PKCζ suppresses the expression of phosphoglycerate dehydrogenase (PHGDH) and phosphoserine aminotransferase (PSAT1): two important enzymes of the pathway [96]. Besides PKCζ, research on the relationship between other PKC isoforms and tumor metabolism is still rare; further studies are expected to figure out the exact mechanism and guide clinical treatment.

## 4. PKC Inhibitors for Cancer Therapy

Since the PKC isoforms are involved in multiple signaling pathways, they are attractive targets for the treatment of various human diseases, including cancers. Different approaches have been proposed to identify the selective regulators of PKC isoforms. For instance, ATP-competitive molecule inhibitors that bind to the ATP site of PKC enzymes, phorbol esters and other derivative activators or inhibitors that bind to the C1 domain for DAG-binding simulation, and peptides that block the interaction between the regulatory region and RACK of PKC enzymes (Table 1).

### 4.1. ATP-Competitive Molecule Inhibitors

ATP-competitive molecule inhibitors are molecules that regulate high-affinity interaction with the ATP binding motif within the catalytic domain of PKC isoforms. Staurosporine is a natural and water-soluble pan-inhibitor of PKC. It has been demonstrated that staurosporine could selectively reverse chronic myeloid leukemia resistance to imatinib by inhibiting PKCα-dependent CDC23 inhibition and arresting the cell cycle in the G2/M phase [97]. Midostaurin (PKC412, CGP41251, N-benzoylstaurosporine), a semi-synthetic derivative of staurosporine, shows stronger inhibition on cPKC than nPKC and is inactive towards PKCζ. Midostaurin exhibited a potent anti-tumor effect on rituximab-resistant Burkitt’s lymphoma (BL) cells by reducing the phosphorylation of PKC and promoting proapoptotic activity. Moreover, the combination of midostaurin and rituximab dramatically improved the survival of mice bearing the resistant BL cells compared to rituximab monotherapy [98]. 7-Hydroxystaurosporine (UCN-01) has a similar preference for PKC as midostaurin and exhibits anticancer activity via dampening cell cycle progression from the G1 to S phase [99]. Several clinical studies indicated that UCN-01 could be administered safely and well-tolerated [100,101]. Sotrastaurin (also known as AEB071) is a broadly selective molecule that blocks the catalytic activity of PKC enzymes, such as PKCα, PKCβ, PKCθ and PKCδ. Targeted PKC inhibition with sotrastaurin exerted an antitumor effect on uveal melanoma cells harboring GNAQ mutations via suppressing PKC/ERK1/2 and PKC/NF-κB pathways [102]. Enzastaurin (also known as LY317615) is an oral selective PKCβ inhibitor that exhibits a synergistic antitumor effect on NSCLC cell growth when combined with pemetrexed [103]. However, the ATP binding motif shares high sequence homology and structural similarity among kinases, which makes it difficult for the inhibitors to achieve the required selectivity. 

### 4.2. Phorbol Esters and Other Derivatives 

Phorbol esters exhibit pharmacological properties via interacting with the C1 domain in PKC, similar to the lipid second messenger DAG, but in a competitive manner [104]. It has been demonstrated that phorbol esters induce cancer cell apoptosis via PKC activation and downregulation of several proto-oncogenes (c-Myc, c-Fos and c-Jun) [105]. Unlike DAG, phorbol esters are not readily metabolized and subsequently fasten PKC in an open and active conformation that is susceptible to ubiquitination, dephosphorylation and degradation. Thus, while phorbol esters induce acute and constitutive activation of PKC, chronic loss or downregulation of all conventional and novel PKC isoforms occurs [106]. Although PKC is the major receptor of phorbol esters, phorbol esters also target other receptors and cause local inflammation. Bryostatin 1 is a macrocyclic lactone that antagonizes the pro-tumor effects of phorbol esters with exceptionally high affinity, leading to significant inhibition of PKCα, PKCε and PKCη, as well as peripheral blood monocytes in patients with advanced malignancies [107]. DAG-lactones are a group of DAG analogs synthesized by a pharmacophore-guided approach, which afford higher affinity for PKC isoforms compared with DAG. HK434 and HK654, two designed compounds of DAG-lactones with highly potent DAG surrogates, are found to induce apoptosis in LNCaP prostate cancer cells by activating PKCα [108]. It is worth noting that the structure of the C1 domain in PKC isoforms is highly conserved, making it arduous to fabricate isoform-selective compounds that target the C1 domain.

### 4.3. Lipid Analogs 

Several lipid analogs have been developed, such as safingol, ilmofosine, perifosine and aurothiomalate, which target activation of PKC at the regulatory domain. Safingol, L-threo-dihydrosphingosine, displayed selective inhibition for PKCα and induced an endonuclease G-mediated cell apoptosis of oral squamous cell carcinoma cells in a caspase-independent manner [109]. Safingol is also a putative inhibitor of sphingosine kinase 1 (SphK), which catalyzes sphingosine 1-phosphate (S1P) and mediates cancer cell growth. A phase I clinical trial showed that a combination of safingol with cisplatin was tolerable in patients with advanced solid tumors [110]. Ilmofosine (BM41.440, 1-hexadecylthio-2-methoxymethyl-rac-glycero-3-phosphocholine), a thioether lysophospholipid derivative of lysophosphocholine, is a PKC inhibitor found to induce G2-arrest and Cdc2 kinase suppression via affecting the formation of Cdc2/cyclin B1 complexes [111]. Ilmosofine exhibits concentration-dependent anticancer activities in various solid tumor models [112,113]. However, no tumor regression occurred in patients with non-small cell bronchogenic carcinoma when using a schedule of continuous infusion for 5 days and a dose of 300 mg/m^2^/day of ilmofosine [114]. Perifosine (D-21226) is a synthetic alkyl phospholipid that demonstrates anticancer effects via inhibition of MAPK, Akt and PKC [115,116,117]. However, according to the trial results of the National Cancer Institute of Canada Clinical Trials Group, perifosine does not show evidence of anticancer activity in patients with previously untreated metastatic or locally advanced soft tissue sarcoma, when given orally on day 1 (900 mg cycle 1, 300 mg cycle 2+) and 2–21 (150 mg daily each cycle) [117]. Aurothiomalate is an anti-rheumatoid gold agent that inhibits PKCλ/ι binding to the adaptor molecule Par6, which regulates Rac1 and induces anchorage-independent growth of NSCLC cells [118]. It also demonstrates a pro-apoptotic effect on advanced prostate cancer and hepatocellular carcinoma [119,120].

### 4.4. Others

Antisense oligonucleotides offer one approach to target genes that regulate cancer cell proliferation, apoptosis, and signaling pathways. Recent clinical trials have confirmed the anti-tumor activity of antisense oligonucleotides with safety and tolerability [121]. Affinitak (ISIS-3521/LY900003) is an antisense oligonucleotide that specifically blocks PKCα [122]. A phase II study demonstrated that ISIS-3521 is effective and safe in patients with relapsed low-grade non-Hodgkin’s lymphoma [123]. However, there is no evidence of any clinical activity or target modulation of ISIS-3521 in patients with metastatic colorectal cancer [124], and no objective responses are observed, even in combination with ISIS-5132 (an antisense inhibitor of Raf-1 kinase) [125].

## 5. Conclusions

It is obvious that PKC isoforms are involved in cancer progression and represent important targets for cancer therapy. However, the biological functions of PKC isoforms are likely to vary across different tumor types. Studies of cancer-associated mutations revealed that the PKC family members are inactivated in cancer, representing a tumor-suppressive function [126]. Based on this theory, strategies of cancer therapy should focus on restoring rather than suppressing PKC function. It also explains why numerous attempts to target PKC isoforms in cancer have yielded very limited success. Clinical trials demonstrated that the use of PKC inhibitors for cancer treatment not only failed but also worsened patient outcomes in some cases. Current efforts of small molecule compounds have been focused on the kinase domain of PKC enzymes. However, PKC enzymes shared a well-conserved kinase domain, which makes it a major challenge to obtain small molecule modulators with high specificity and sensitivity toward a specific PKC isoform. Despite this, PKC isoforms are promising anti-tumor targets and additional efforts are needed to improve the efficacy and selectivity of PKC isoform inhibitors.

## Figures and Tables

**Figure 1 cancers-14-01104-f001:**
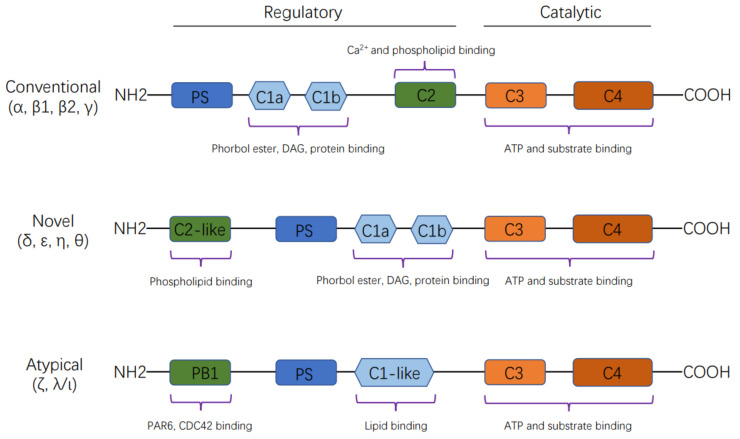
Schematic representation of the structure of protein kinase C (PKC) isoforms. The PKC family members are classified into three subfamilies according to their cofactor dependence: (i) conventional or classical PKC isozymes (cPKCs), including PKCα, PKCβ1, PKCβ2 and PKCγ, which are calcium-dependent and activated by both diacylglycerol (DAG) and phosphatidylserine (PS); (ii) novel PKC isozymes (nPKCs), including PKCδ, PKCε, PKCη and PKCθ, which are calcium-independent and regulated by DAG and PS; and (iii) atypical PKC isozymes (aPKCs), including PKCζ and PKCλ/ι, which are calcium-independent and do not require DAG for activation, although PS can regulate their activity.

**Figure 2 cancers-14-01104-f002:**
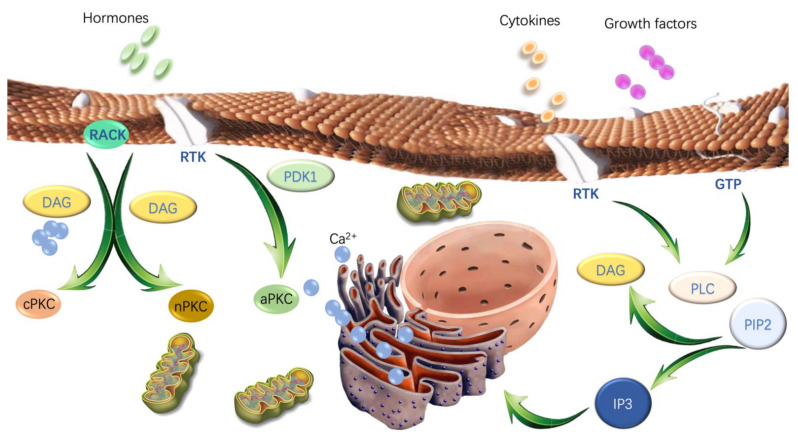
Schematic representation of the activation of protein kinase C (PKC) isoforms. Before activation, PKC isoforms need to be maturated by 3-phosphoinositide-dependent protein kinase-1 (PDK1). Under physiological conditions, PKC can be activated by interacting with extracellular agonists, such as hormones, cytokines and growth factors, which sequentially translocates to different subcellular compartments, thus conferring specific substrate phosphorylation and distinct cellular functions. The translocation of PKC is regulated by anchoring/scaffolding proteins to ensure proper spatio-temporal distribution of PKC isoforms. Receptors for activated C kinase (RACKs) are intracellular scaffolding proteins that bind to individual PKC isoforms following their activation and provide anchorage of each isoform next to its physiological substrates. Subsequently, the translocated PKC phosphorylates specific substrates, such as membrane-associated phospholipase C (PLC), which hydrolyzes the membrane phospholipid phosphatidylinositol 4,5-bisphosphate (PIP2) to generate diacylglycerol (DAG) and inositol 1,4,5-trisphosphate (IP3). IP3 then interacts with the inositol trisphosphate receptor (InsP3R) and triggers the rapid release of Ca^2+^ from the intracellular store to the endoplasmic reticulum (ER). DAG is phosphorylated by diacylglycerol kinases (DGK) and converts to phosphatidic acid (PA) in a reaction that terminates PKC-regulated signals.

**Table 1 cancers-14-01104-t001:** Inhibitors of PKC isoforms.

Structure	Name	Specificity	Inhibition Mechanism
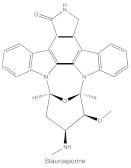	Staurosporine	Pan-PKCs	competitive interfere with the ATP binding site
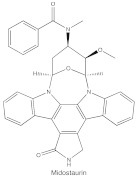	Midostaurin (PKC412, CGP41251, N-benzoylstaurosporine)	cPKCs, nPKCs	competitive interfere with the ATP binding site
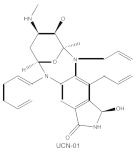	7-Hydroxystaurosporine (UCN-01)	cPKCs, nPKCs	competitive interfere with the ATP binding site
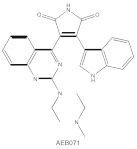	Sotrastaurin (AEB071)	PKCα, β, θ, δ	competitive interfere with the ATP binding site
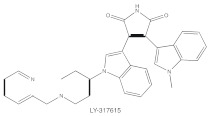	Enzastaurin (LY317615)	PKCβ	competitive interfere with the ATP binding site
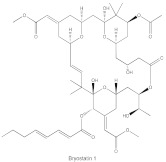	Bryostatin 1	PKCα, PKCε, and PKCη	competitive interfere with the phorbol ester binding site

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
