# Peer review of "Targeting Protein Kinase C for Cancer Therapy"

_cancers, 2022, doi:10.3390/cancers14051104_

Round 1
Reviewer 1 Report
Dear authors,
Congratulations on your nice and interesting review paper. You can find my comment below:
- In section 3.2, the role of PKC in anticancer treatment does not describe very well. Please include detailed examples, since it is very important for the readers.
- Please include 1 or 2 figures with chemical structures of mentioned molecules that interact with PKCs (inhibitors mainly).
- Can you relate the inhibition mechanism regarding the structure of the inhibitors with the structure of active PKCs centers (hydrogen bonds, van der Waals interactions, polarity e.t.c.)?
- Please revise your manuscript for grammatical and spelling errors.
Best regards,
Reviewer 2 Report
Authors review PKC isoforms, structure and relation to cancer (therapy) Overall, the manuscripts is well written (with a lot of spelling errors, see below). Some aspects are highlighted more extensively compared to others.
Major comments:
The normal function of PKCs is only briefly addressed in the introduction. Yet, its crucial to understand its implications in cancer related processes. Please elaborate more on the processes and pathways PKC are involved in.
In all sections, there is a lot of detail about the involvement, expression, mutations in/of PKC in process X or Y. However, the manuscript either lacks detail on " how" PKCs are involved in the processes discussed or the section is organized in such a way that the information regarding the "how" is scattered throughout the section. This makes the review difficult to follow.
Minor comments
There are numerous spelling and grammar errors throughout the text. For instance: in the first three lines of text there are already three errors:
Simple Summary: The protein kinase C (PKC) family belongs to serine-threonine kinases and consisted of several subtypes. Numerous researches demonstrate that PKCs are critical players in carcinogenesis. Interestedly
Please use a language service or ask a native speaker to read and edit the manuscript
Figure 2: the text and text objects are of a far less resolution as the other figure objects and seem to be copy-pasted in a really elegant and detailed figure. I would suggest to improve the figure objects and text.
Round 2
Reviewer 2 Report
The manuscripy has improved a lot. i have no further suggestions
